# Molecular Aspects of Drug-Induced Gingival Overgrowth: An In Vitro Study on Amlodipine and Gingival Fibroblasts

**DOI:** 10.3390/ijms20082047

**Published:** 2019-04-25

**Authors:** Dorina Lauritano, Alberta Lucchese, Dario Di Stasio, Fedora Della Vella, Francesca Cura, Annalisa Palmieri, Francesco Carinci

**Affiliations:** 1Department of Medicine and Surgery, Centre of Neuroscience of Milan, University of Milano-Bicocca, 20126 Milan, Italy; 2Multidisciplinary Department of Medical and Dental Specialties, University of Campania- Luigi Vanvitelli, 80138 Naples, Italy; alberta.lucchese@unicampania.it (A.L.); dario.distasio@unicampania.it (D.D.S.); 3Interdisciplinary Department of Medicine, University of Bari, 70121 Bari, Italy; fedora.dellavella@uniba.it; 4Department of Experimental, Diagnostic and Specialty Medicine, University of Bologna, via Belmoro 8, 40126 Bologna, Italy; cura.francesca@tiscali.it (F.C.); annalisa.palmieri@unife.it (A.P.); 5Department of Morphology, Surgery and Experimental Medicine, University of Ferrara, 44121 Ferrara, Italy; crc@unife.it

**Keywords:** Gingival overgrowth, gene expression, drugs, amlodipine

## Abstract

Gingival overgrowth is a serious side effect that accompanies the use of amlodipine. Several conflicting theories have been proposed to explain the fibroblast’s function in gingival overgrowth. To determine whether amlodipine alters the fibrotic response, we investigated its effects on treated gingival fibroblast gene expression as compared with untreated cells. Materials and Methods: Fibroblasts from ATCC^®^ Cell Lines were incubated with amlodipine. The gene expression levels of 12 genes belonging to the “Extracellular Matrix and Adhesion Molecules” pathway was investigated in treated fibroblasts cell culture, as compared with untreated cells, by real time PCR. Results: Most of the significant genes were up-regulated. (*CTNND2*, *COL4A1*, *ITGA2*, *ITGA7*, *MMP10*, *MMP11*, *MMP12*, *MMP26*) except for *COL7A1*, *LAMB1*, *MMP8*, and *MMP16*, which were down-regulated. Conclusion: These results seem to demonstrate that amlodipine has an effect on the extracellular matrix of gingival fibroblast. In the future, it would be interesting to understand the possible effect of the drug on fibroblasts of patients with amlodipine-induced gingival hyperplasia.

## 1. Introduction

Gingival overgrowth may have multiple causes, however drugs assumption is the most common [1,2]. In addiction, drug-induced gingival overgrowth may be associated with a patient’s genetic predisposition [3,4].

The three main classes of drugs inducing gingival overgrowth are anticonvulsants, immunosuppressive, and antihypertensive agents [3,5,6,7]. The first report regarding gingival overgrowth by the administration of amlodipine was published in 1994 [8]. Subsequently, other authors reported the onset of gingival overgrowth as a side effect in patients who received 10 mg per day of amlodipine within two months [9]. Gingival overgrowth manifests as a side effect within one to three months after amlodipine administration [7,10]. Amlodipine shows a pharmacological profile, as follows: long-acting dihydropyridine; coronary and peripheral arterial vasodilatation; headaches, facial flushing, dizziness, and oedema. The main oral side effect that is induced by amlodipine is gingival overgrowth [11,12].

Gingival overgrowth that is induced by amlodipine (GOIA) must be stimulated by a threshold concentration of amlodipine [13]. However the severity of GOIA is supposed to be related to the concentration of amlodipine in oral fluids [13]. The mean dose of amlodipine reported to cause GOIA in most of the subjects is 5 mg/day. Therefore, it may be suggested that the dosage and duration of amlodipine may have an impact on GOIA [13]. Usually, the gingival manifestations of GOIA appear within the first three months of the drug administration [14]. A longer duration of therapy may increase the exposure of cells to amlodipine, and this may modify the apoptosis of cells, resulting in hyperplasia [14]. Genetic susceptibility is one the factors influencing the severity of GOIA; in fact multidrug resistant (*MDR1*) gene polymorphisms is supposed to alter the inflammatory response to the amlodipine [15].

Poor plaque control is another risk factor in developing of GOIA. In fact, GOIA hampered routine oral hygiene measures. Additionally, GOIA can favour the accumulation of bacterial plaque, which in turn determines gingival inflammation, causing gingival overgrowth [14]. Amlodipoine, in presence of proinflammatory cytokines (for example TNF-α), may favour the inhibition of apoptosis of human gingival fibroblasts, thus promoting hyperplasia [16].

### 1.1. Genetic Factors

Evidence suggests that genetic factors, along with patient susceptibility, may play an important role in the pathogenesis of GOIA [15]. A genetic predisposition can influence a number of factors in the drug interactions, cells, and plaque-induced inflammation. These include the functional heterogeneity of gingival fibroblasts, the collagenolytic activity, the drug-receptor binding, the drug metabolism, collagen synthesis, and many other factors.

Since most types of pharmacological agents that are implicated in GOIA can have negative effects on the flow of calcium ions across cell membranes, it has been postulated that these agents can interfere with the synthesis and function of the collagenase [17]. This mechanism of action has been demonstrated for Cyclosporine. In fact, a recent study in vitro have shown that human gingival fibroblasts that were treated with relevant doses of Cyclosporine show significantly reduced levels of secretion of MMP-1 and MMP-3; these reduced levels can contribute to the accumulation of components of the extracellular matrix [18]. An animal study that showed low mRNA levels of collagenase in situ further supported these results, accompanied by a decrease in phagocytosis and degradation of collagen [19]. The genetic predisposition to GOIA has been studied for cyclosporine, while, for what we know, not yet for amlodipine.

### 1.2. Objective

To determine whether amlodipine can alter the matrix deposition, we investigated its effects on treated gingival fibroblast gene expression as compared with untreated cells.

## 2. Results

The PrestoBlue™ cell viability test was conducted to determine the optimal concentration of amlodipine to be used for cell treatment that did not significantly affect cell viability. Based on this test, the concentration used for the treatment was 1000 ng/mL.

The gene expression profile of 12 genes belonging to the “Extracellular Matrix and Adhesion Molecules” pathway was analysed using Real time PCR (Figure 1). Table 1 reported the list gene and their fold change.

Bold fonts indicate significant variation of gene expression level: fold change ≥ 2 and *p* value ≤ 0.05 for up-regulated genes, and fold change ≤ 0.5 and *p* value ≤ 0.05 for the significantly down-regulated genes.

Among the up-regulated genes, there was *CTNND2*, which code for the cell adhesion protein Catenin Delta 2. Other up-regulated gens were the transmembrane receptor *ITGA2* and *ITGA7* and the basement membrane constituent *LAMB1*. Most of the extracellular matrix proteases were up-regulated (*MMP10*, *MMP11*, *MMP12*, *MMP26*), except for *MMP8* and *MMP16*, which were down-regulated. Other genes that were significantly deregulated genes following the treatment with Amlodipine were *COL7A1*, which was down-regulated and the *COL4A1* that was up-regulated. In Figure 1 the significantly expression levels of the genes up- and down-regulated in fibroblast cells treated with amlodipine were represented.

## 3. Discussion

The prevalence of GOIA might be as high as 38%, and it is 3.3 times more common in men than in women [20,21]. The pathogenesis may be different for different drugs, even if the oral manifestations of gingival overgrowth are similar. GOIA starts as an enlargement of the interdental papilla of keratinized portions of the gingiva, and it is characterized by an increase in the connective tissue component. Bacteria accumulation appears to be an important uncomfortable effect of GOIA. GOIA may impair oral hygiene and lead to increased oral infections. Oral infection itself is a cause of gingival overgrowth [22]. In addiction, oral infection can potentially impair systemic health and could possibly compromise the general health of patients [22].

The mechanism of action of GOIA is still unknown, however it may be a consequence of the interaction between gingival fibroblasts, cellular and biochemical mediators of inflammation, and drug metabolites [22,23].

Gingival overgrowth is documented more frequently after intake of with phenytoin and rarely with others antihypertensive [24]. Furthermore, poor oral hygiene is indicated as an important risk factor for the expression of GOIA [25,26].

Cross-sectional studies have reported the relationship between bacterial plaque and GOIA. In fact, as previously reported, GOIA can favour the accumulation of bacterial plaque, which in turn determines gingival inflammation, causing gingival overgrowth [26]. The underlying mechanism of GOIA still remains to be fully understood, however two main inflammatory and non-inflammatory pathways have already been suggested [27,28,29]. One hypothesis may be that amlodipine may induce the alteration of collagenase activity as a consequence of decreased uptake of folic acid, blockage of aldosterone synthesis in adrenal cortex, and consequent feedback increase in the adrenocorticotropic hormone level and the up-regulation of keratinocyte growth factor [30]. Besides, inflammation may be a consequence of the toxic effect of amlodipine in periodontal pocket associated with C pathogens, leading to the up-regulation of several cytokine factors, such as transforming growth factor-beta 1 (TGF-β1) [9]. Another pathogenic mechanism of GOIA is focusing on the effects of amlodipine on gingival fibroblast metabolism and genetic predisposition. In fact, only a subgroup of patients that were treated with this amlodipine will develop GOIA, so it has been hypothesized that these individuals show an abnormal susceptibility to the drug. In fact, elevated levels of protein synthesis, most of which is collagen, characterize the fibroblast of GOIA in these patients. Treatment of GOIA is generally targeted on drug substitution and preventive protocols [12,31]. Surgical intervention is recommended when these measures fail to cause the resolution of GOIA. These treatment modalities, although effective, do not necessarily prevent the recurrence of the lesions [12]. Surgery for treatment of GOIA must be carefully assessed and it is normally performed for cosmetic/aesthetic needs before any functional consequences are present [1,32]. Most reports of GOIA have required surgical intervention [25].

To our knowledge, our study is the first one analysing the effect of amlodipine on genes that belong to the “Extracellular Matrix and Adhesion Molecules” pathway. In this study, gingival fibroblasts were treated for 24 h with 1000 ng/mL of amlodipine. The gene expression profile of 12 genes that belong to the “Extracellular Matrix and Adhesion Molecules” pathway was analysed. Most of the significant genes were up-regulated. (*CTNND2*, *COL4A1*, *ITGA2*, *ITGA7*, *MMP10*, *MMP11*, *MMP12*, *MMP26*), except for *COL7A1*, *LAMB1*, *MMP8*, and *MMP16*, which were down-regulated. These proteins preferentially induce extra cellular matrix deposition. This study demonstrated that, in human gingival fibroblasts that were cultivated in vitro, amlodipine could promote the activities of genes belonging to the “fibroblast matrix and receptors”.

It might be part of the underlying reason for the persistent overgrowth of gingiva that was seen when bacterial plaque and local inflammation are present during amlodipine therapy. In fact, GOIA does not allow patient to maintain a good oral hygiene, and this is the reason why GOIA always determines the presence of bacterial plaque and inflammation, which in turn determines gingival overgrowth.

The data presented here suggest that amlodipine may contribute to an extracellular matrix deposition of human gingival fibroblasts inducing gingival overgrowth.

## 4. Materials and Methods

### 4.1. Primary Human Fibroblast Cells Culture

We used cells from ATCC^®^ Cell Lines. The cryopreserved cells at the second passage were cultured in 75 cm^2^ culture flasks containing DMEM medium (Sigma Aldrich, Inc., St Louis, Mo, USA) supplemented with 20% fetal calf serum, antibiotics (Penicillin 100U/ml and Streptomycin 100 micrograms/ml-Sigma Aldrich, Inc., St Louis, Mo, USA).

Cell cultures were replicated for subsequent experiments and maintained in a water saturated atmosphere at 37 °C and 5% CO_2_.

### 4.2. Cell Viability Test

A stock solution of amlodipine 1 mg/mL was prepared. Further dilutions were made with the culture medium to the desired concentrations just before use. The cell lines were seeded into 96-well plates at a density of 104 cells per well containing 100 µL of cell culture medium and incubated for 24 h to allow cell adherence. Serial dilution of amlodipine (5000 ng/mL, 2000 ng/mL, 1000 ng/mL, 500 ng/mL, 100 ng/mL) was added (three wells for each concentration). The cell culture medium alone was used negative control.

After 24 h of incubation, cell viability was measured while using PrestoBlue™ Reagent Protocol (Invitrogen, Carlsbad, CA, USA) according to the manufacturer’s instructions. Briefly, the PrestoBlue™ solution (10 µL) was added into each well containing 90 µL of treatment solution. The plates were then placed back into the incubator for 1 h, after which absorbance was measured at wavelengths of 570 nm excitation and 620 nm emission by an automated microplate reader (Sunrise™, Tecan Trading AG, Switzerland). Comparing the average absorbance in drug treated wells with average absorbance in control wells exposed to vehicle alone determined the percentage of viable cells.

### 4.3. Cell Treatment

Gingival fibroblasts were seeded at a density of 1.0 × 10^5^ cells/ml into 9 cm^2^ (3 mL) wells and then subjected to serum starvation for 16 hours at 37 °C. Cells were treated with 1000 ng/mL amlodipine solution for 24 h. This solution was obtained in DMEM that was supplemented with 2% FBS, antibiotics and aminoacids. Cell medium alone was used as control negative. The cells were maintained in a humidified atmosphere of 5% CO_2_ at 37 °C. After the end of the exposure time, the cells were trypsinized and processed for RNA extraction.

### 4.4. RNA Isolation, Reverse Transcription and Quantitative Real-Time RT-PCR

Total RNA was isolated from cultured cells using GenElute mammalian total RNA purification miniprep kit (Sigma-Aldrich, Inc., St Louis, Mo, USA), according to manufacturer’s instructions. Pure RNA was quantified at NanoDrop 2000 spectrophotometer (Thermo Scientific, Waltham, Massachusetts, USA).

cDNA synthesis was performed, starting from 500 ng of total RNA, using PrimeScript RT Master Mix (Takara Bio Inc, Kusatsu, Japan). The reaction was incubated at 37 °C for 15 min. and inactivated by heating at 70 °C for 10 s.

cDNA was amplified by Real Time Quantitative PCR using the ViiA™ 7 System (Applied Biosystems, Foster City, CA, USA).

All of the PCR reactions were performed in a 20 µL volume. Each reaction contained 10 µL of 2x qPCRBIO SYGreen Mix Lo-ROX (Pcrbiosystems, London, UK), 400 nM concentration of each primer, and cDNA. Table 2 reported the sequences of the primer that was used in the reaction.

Custom primers belonging to the “Extracellular Matrix and Adhesion Molecules” pathway were purchased from Sigma Aldrich. All of the experiments were performed, including non-template controls to exclude reagents contamination. PCR was performed, including two analytical replicates.

The amplification profile was initiated by 10 min. incubation at 95 °C, followed by two-step amplification of 15 s at 95 °C and 60 s at 60 °C for 40 cycles. As final step, a melt curve dissociation analysis was performed.

### 4.5. Statistical Analysis

The gene expression levels were normalized to the expression of the reference gene (RPL13) and they were expressed as fold changes relative to the expression of the untreated cells. Quantification was done with the delta/delta Ct calculation method.

## 5. Conclusions

In this study, most of the significantly genes belonging to the “extracellular matrix proteases” pathway were up-regulated. These results seem to indicate that amlodipine has an effect on the modulation of fibrosis response in gingival fibroblasts, up-regulating extracellular matrix proteases, and favouring the deposition of fibrotic tissue. More explanatory results could probably be obtained by using gingival fibroblasts in which the use of amlodipine seems to aggravate the fibrotic response and the gingival overgrowth.

GOIA is no longer a rare occurrence. From one side, plaque accumulation is an inevitable consequence of GOIA, and from the other, it favours gingival inflammation and overgrowth. The duration of therapy, dosage, and individual genetic susceptibility are considered important risk factors for the development of GOIA. Amlodipine is a widely used drug for the treatment of hypertension and angina, so it is very important that doctors inform patients about side effects, such as GOIA, and about the importance of preventive protocols. Dentists should be able to identify the changes in the oral cavity that are related to the general health of their patients. The patients must be informed of the tendency of certain drugs to cause gingival overgrowth and the associated oral changes and the importance of effective oral hygiene.

Combination therapy consisting of surgical and non-surgical periodontal therapy with drug substitution is the most reliable method in the management of GOIA.

## Figures and Tables

**Figure 1 ijms-20-02047-f001:**
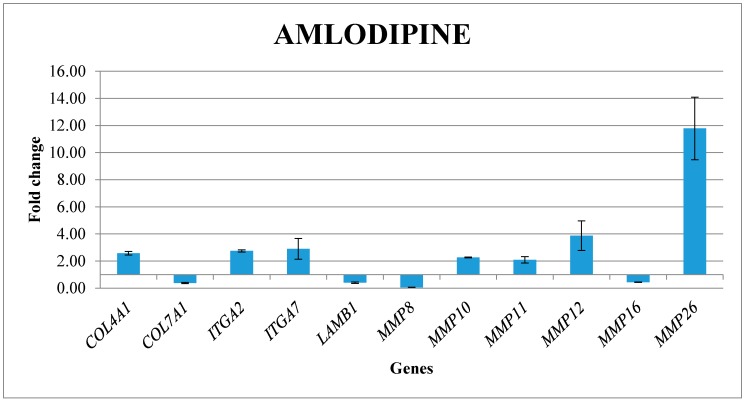
Gene expression profile of fibroblast treated with Amlodipine 1000 ng/mL.

**Table 1 ijms-20-02047-t001:** Significant gene expression levels after 24 h treatment with Amlodipine, as compared with untreated cells.

Gene	Fold Change	SD (+/−)	Gene Function
*CTNND2*	**2.29**	0.03	Cell Adhesion Molecule
*COL4A1*	**2.57**	0.13	Collagens & Extracellular Matrix Structural constituent
*COL7A1*	**0.38**	0.04	Collagens & Extracellular Matrix Structural constituent
*ITGA2*	**2.75**	0.08	Transmembrane Receptor
*ITGA7*	**2.90**	0.77	Transmembrane Receptor
*LAMB1*	**0.41**	0.05	Basement Membrane Constituent
*MMP8*	**0.06**	0.01	Extracellular Matrix Protease
*MMP10*	**2.27**	0.03	Extracellular Matrix Protease
*MMP11*	**2.09**	0.24	Extracellular Matrix Protease
*MMP12*	**3.88**	1.09	Extracellular Matrix Protease
*MMP16*	**0.44**	0.02	Extracellular Matrix Protease
*MMP26*	**11.78**	2.31	Extracellular Matrix Protease

**Table 2 ijms-20-02047-t002:** Primers sequences of SYBR^®^ Green assay.

Gene Name	Forward Sequence 5′ > 3′	Reverse Sequence 5′ > 3′
*CTNND2*	AGAGAATTTGGATGGAGAGAC	TTGTTGTCTCCAAAACAGAG
*COL4A1*	AAAGGGAGATCAAGGGATAG	TCACCTTTTTCTCCAGGTAG
*COL7A1*	ATGACCTTGGCATTATCTTG	TGAATATGTCACCTCTCAAGG
*ITGA2*	GGTGGGGTTAATTCAGTATG	ATATTGGGATGTCTGGGATG
*ITGA7*	CATGAACAATTTGGGTTCTG	GCCCTTCCAATTATAGGTTC
*LAMB1*	GTGTGTATAGATACTTCGCC	AAAGCACGAAATATCACCTC
*MMP8*	AAGTTGATGCAGTTTTCCAG	CTGAACTTCCCTTCAACATTC
*MMP10*	AGCGGACAAATACTGGAG	GTGATGATCCACTGAAGAAG
*MMP11*	GATAGACACCAATGAGATTGC	TTTGAAGAAAAAGAGCTCGC
*MMP12*	AGGTATGATGAAAGGAGACAG	AGGTATGATGAAAGGAGAACAG
*MMP16*	ACCCTCATGACTTGATAACC	TCTGTCTCCCTTGAAGAAATAG
*MMP26*	AAGGATCCAGCATTTGTATG	CTTTGATCCTCCAATAAACTCC
*RPL13*	AAAGCGGATGGTGGTTCCT	GCCCCAGATAGGCAAACTTTC

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
