# Peer review of "Molecular Aspects of Drug-Induced Gingival Overgrowth: An In Vitro Study on Amlodipine and Gingival Fibroblasts"

_ijms, 2019, doi:10.3390/ijms20082047_

Round 1

Reviewer 1 Report

The authors have examined the molecular aspects of amlodipine induced gingival overgrowth. I have the following comments

Major Comments

In discussion the authors have claimed the upregulation of many ECM and adhesion molecules in their study. Though they have claimed that these proteins induce ECM deposition, there are no references cited to support their claim. In the next line the authors have mentioned ‘co-cultures’, but in the materials and methods the authors have mentioned using only fibroblasts for their study. The authors need to clarify the use of the term ‘co cultures’.

The authors have suggested that the overexpression of the above mentioned genes might be the reason for persistent overgrowth when there is bacterial plaque and local inflammation during therapy. Could the authors clarify more on this. How the authors have correlated between plaque, inflammation, amlodipine therapy and increased gene expression in their study.

The authors have claimed that amlodipine was actively involved in the migration of gingival fibroblasts. What experiments were performed to assess this? The authors have further claimed inhibition of migration through inhibition of MMP 8 &16. Is this an observation in their study or are there any references to their statement. In the next line authors have made a statement that their findings reveal a therapeutic potential of amlodipine on progression of gingival hyperplasia. Could the authors clarify more upon this. Since the authors claim that change in gene expression due to amlodipine may cause gingival overgrowth. These statements by the authors are contradictory.

Minor comments:

In abstract: what is the purpose of citing Seymour et al. this is not introduction. please remove. 

In materials and methods the authors need to mention more about the details of cell lines used. Were they gingival fibroblasts?

In results the authors have generically mentioned all genes upregulated as ECM proteases which all are not.

The authors need to do a thorough grammar and spell check. In the areas where numbers have been used the use of superscript needs to be precise. For example, where the authors have mentioned the seeding density of cells. The authors have mentioned use of ‘bold fonts’. The manuscript doesn’t show any of this.

Author Response

Milan 18th April 2019

            Dear editor,

            many thanks for the insightful comments and suggestions of the referees.

            We have made corresponding revision according to their advice. Words in red are the changes we have made in the text. The language of the manuscript has also been extensively

revised by a professional English language science editing service and all authors of

this article have seen and approved the changes.

            The revisions are as follows:

            Major Comments

- In discussion the authors have claimed the upregulation of many ECM and adhesion molecules in their study. Though they have claimed that these proteins induce ECM deposition, there are no references cited to support their claim. In the next line the authors have mentioned ‘co-cultures’, but in the materials and methods the authors have mentioned using only fibroblasts for their study. The authors need to clarify the use of the term ‘co cultures’.

Though they have claimed that these proteins induce ECM deposition, there are no references cited to support their claim.

We added the reference These proteins preferentially induce extra cellular matrix deposition30

In the next line the authors have mentioned ‘co-cultures’, but in the materials and methods the authors have mentioned using only fibroblasts for their study. The authors need to clarify the use of the term ‘co cultures’.

The term co-colture was substituited with “In vitro culture”

The authors have suggested that the overexpression of the above mentioned genes might be the reason for persistent overgrowth when there is bacterial plaque and local inflammation during therapy. Could the authors clarify more on this. How the authors have correlated between plaque, inflammation, amlodipine therapy and increased gene expression in their study.

We modified the sentence as follow:

It might be part of the underlying reason for the persistent overgrowth of gingiva seen when bacterial plaque and local inflammation are present during amlodipine therapy. In fact GOIA don’t allow patient to maintain a good oral hygiene, and this is the reason why GOIA always determines the presence of bacterial plaque and inflammation, which in turn determines itself gingival enlargement.

-            The authors have claimed that amlodipine was actively involved in the migration of gingival fibroblasts. What experiments were performed to assess this? The authors have further claimed inhibition of migration through inhibition of MMP 8 &16. Is this an observation in their study or are there any references to their statement. In the next line authors have made a statement that their findings reveal a therapeutic potential of amlodipine on progression of gingival hyperplasia. Could the authors clarify more upon this. Since the authors claim that change in gene expression due to amlodipine may cause gingival overgrowth. These statements by the authors are contradictory.

We removed this sentence.

Minor comments:

- In abstract: what is the purpose of citing Seymour et al. this is not introduction. please remove. 

Seymour et al. citation was removed from the abstract

- In materials and methods the authors need to mention more about the details of cell lines used. Were they gingival fibroblasts?

More details of cell lines used were added to the materials and methods section: “Human oral fibroblast derived from the gingival tissue of 60 years female were purchased from ATCC® Cell Lines. Cryopreserved cells at the second passage were cultured in 75 cm2 culture flasks containing DMEM medium…..”

- In results the authors have generically mentioned all genes upregulated as ECM proteases which all are not.

In the results section the deregulated genes were better detailed: “Among the up-regulated genes there was CTNND2 that code for the cell adhesion protein Catenin Delta 2. Other up-regulated gens were the transmembrane receptor ITGA2 and ITGA7 and the basement membrane constituent LAMB1. Most of the extracellular matrix proteases were up-regulated (MMP10, MMP11, MMP12, MMP26) except for MMP8 and MMP16 which were down-regulated. Other genes significantly deregulated genes following the treatment with Amlodipine were COL7A1 that was down-regulated and the COL4A1 that was up-regulated.”

The authors need to do a thorough grammar and spell check. In the areas where numbers have been used the use of superscript needs to be precise. For example, where the authors have mentioned the seeding density of cells. The authors have mentioned use of ‘bold fonts’. The manuscript doesn’t show any of this.

Superscript number were checked

The bold font was added in the table as indicated in the text of the manuscript.

Thank you for receiving our manuscript and considering it for pubblication.

We appreciate your time and look forward to your response.

Yours sincerely,

Dorina Lauritano

Reviewer 2 Report

In their manuscript titled “Molecular aspects of drug-induced gingival overgrowth: an in vitro study on amlodipine and gingival fibroblasts”, the authors demonstrated the effect of drug treatment on expression profile of 12 genes of extracellular matrix and adhesion molecules pathway. This is an interesting finding.

Following are my comments if they can be addressed;

There are many grammatical mistakes throughout the manuscipt. Please make sure you address all of them.

Line 32, “In the future, it would be………………………..affected by gingival hyperplasia”. I think it may be more interesting to check the gene expression profiles in patients with amlodipine-induced gingival hyperplasia, i.e., in gingival tissues already affected by amlodipine – ex vivo studies.

Line 37, please explain or rephrase the sentence. It is very confusing.

Line 59-60, “Longer duration of therapy………………….viability and growth”. Cite/reference this sentence.

Line 98, “After 15 days…………….24h of incubation”. I think you are referring to the cells grown in culture. This should not be written as “pieces of gingival tissue”. They are cells not tissue.

Line 134, list sequences for all primers used for the 12 genes.

Line 154, no bold fonts are seen here. Please check

Line 159, the authors state expression of COL7A1 gene is upregulated, but Fig 1 shows that it is downregulated. Please explain.

Will appreciate if the authors can expand a little more on the results rather than just showing in table 1.

 Line 204, change “up-regulated” to “down-regulated”. Check expression for COL7A1.

 Line 217, not all genes were up-regulated. Authors should state that most of the significant genes were up-regulated.

Author Response

Milan 18th April 2019

            Dear editor,

            many thanks for the insightful comments and suggestions of the referees.

            We have made corresponding revision according to their advice. Words in red are the changes we have made in the text. The language of the manuscript has also been extensively

revised by a professional English language science editing service and all authors of

this article have seen and approved the changes.

            The revisions are as follows:

In their manuscript titled “Molecular aspects of drug-induced gingival overgrowth: an in vitro study on amlodipine and gingival fibroblasts”, the authors demonstrated the effect of drug treatment on expression profile of 12 genes of extracellular matrix and adhesion molecules pathway. This is an interesting finding.

Following are my comments if they can be addressed;

- There are many grammatical mistakes throughout the manuscipt. Please make sure you address all of them.

 The language of the manuscript has also been extensively revised by a professional English language science editing service and all authors of this article have seen and approved the changes.

- Line 32, “In the future, it would be………………………..affected by gingival hyperplasia”. I think it may be more interesting to check the gene expression profiles in patients with amlodipine-induced gingival hyperplasia, i.e., in gingival tissues already affected by amlodipine – ex vivo studies.

The sentence was reformulated as: “In the future, it would be interesting understand, the possible effect of the drug on fibroblasts of patients with amlodipine-induced gingival hyperplasia”

- Line 37, please explain or rephrase the sentence. It is very confusing.

The sentence was reformulted as: “Gingival overgrowth, may be induced by a lot of drugs, used mainly for non-dental treatment for which the gingival tissue is not the intended target organ

- Line 59-60, “Longer duration of therapy………………….viability and growth”. Cite/reference this sentence. ?

Reference was added

- Line 98, “After 15 days…………….24h of incubation”. I think you are referring to the cells grown in culture. This should not be written as “pieces of gingival tissue”. They are cells not tissue.

The sentence was corrected as: “Cell cultures were replicated for subsequent experiments and maintained in water saturated atmosphere at 37 °C and 5% CO2”.

- Line 134, list sequences for all primers used for the 12 genes.

All the primer sequences were listed in table 1. Table reporting the gene expression level of genes was renamed as Table 2

 - Line 154, no bold fonts are seen here. Please check

The bold font was added in the table as indicated in the text of the manuscript

- Line 159, the authors state expression of COL7A1 gene is upregulated, but Fig 1 shows that it is downregulated. Please explain.

 In the results section the deregulated genes were better detailed and mistakes corrected: “Among the up-regulated genes there was CTNND2 that code for the cell adhesion protein Catenin Delta 2. Other up-regulated gens were the transmembrane receptor ITGA2 and ITGA7 and the basement membrane constituent LAMB1. Most of the extracellular matrix proteases were up-regulated (MMP10, MMP11, MMP12, MMP26) except for MMP8 and MMP16 which were down-regulated. Other genes significantly deregulated genes following the treatment with Amlodipine were COL7A1 that was down-regulated and the COL4A1 that was up-regulated.”

- Will appreciate if the authors can expand a little more on the results rather than just showing in table 1.

We added a section explaining results.

- Line 204, change “up-regulated” to “down-regulated”. Check expression for COL7A1.

The sentence was corrected as: Most of the significant genes were up-regulated. (CTNND2, COL4A1, ITGA2, ITGA7, MMP10, MMP11, MMP12, MMP26) except for COL7A1, LAMB1, MMP8 and MMP16, which were down-regulated.

- Line 217, not all genes were up-regulated. Authors should state that most of the significant genes were up-regulated.

The sentence was corrected as:  In this study, most of the significantly genes belonging to the “extracellular matrix proteases” pathway were up-regulated

Thank you for receiving our manuscript and considering it for pubblication.

We appreciate your time and look forward to your response.

Yours sincerely,

Dorina Lauritano

Round 2

Reviewer 1 Report

thanks for your response.